# HIV-1 Transcription Inhibitor 1E7-03 Restores LPS-Induced Alteration of Lung Leukocytes’ Infiltration Dynamics and Resolves Inflammation in HIV Transgenic Mice

**DOI:** 10.3390/v12020204

**Published:** 2020-02-12

**Authors:** Marina Jerebtsova, Asrar Ahmad, Xiaomei Niu, Ornela Rutagarama, Sergei Nekhai

**Affiliations:** 1Department of Microbiology, College of Medicine, Howard University, Washington, DC 20059, USA; orutagarama@gmail.com; 2Center for Sickle Cell Disease, Howard University, Washington, DC 20059, USA; asrar.ahmad@howard.edu (A.A.); xniu@howard.edu (X.N.); 3Department of Medicine, Howard University, Washington, DC 20059, USA

**Keywords:** HIV-transgenic mice, non-infectious lung disease, transcriptional inhibitor, leukocyte dynamics

## Abstract

Human immunodeficiency virus (HIV)-infected individuals treated with anti-retroviral therapy often develop chronic non-infectious lung disease. To determine the mechanism of HIV-1-associated lung disease we evaluated the dynamics of lung leukocytes in HIV-1 transgenic (Tg) mice with integrated HIV-1 provirus. In HIV-Tg mice, lipopolysacharide (LPS) induced significantly higher levels of neutrophil infiltration in the lungs compared to wild-type (WT) mice. In WT mice, the initial neutrophil infiltration was followed by macrophage infiltration and fast resolution of leukocytes infiltration. In HIV-Tg mice, resolution of lung infiltration by both neutrophils and macrophages was significantly delayed, with macrophages accumulating in the lumen of lung capillaries resulting in a 45% higher rate of mortality. Trans-endothelial migration of HIV-Tg macrophages was significantly reduced in vitro and this reduction correlated with lower HIV-1 gene expression. HIV-1 transcription inhibitor, 1E7-03, enhanced trans-endothelial migration of HIV-Tg macrophages in vitro, decreased lung neutrophil infiltration in vivo, and increased lung macrophage levels in HIV-Tg mice. Moreover, 1E7-03 reduced levels of inflammatory IL-6 cytokine, improved bleeding score and decreased lung injury. Together this indicates that inhibitors of HIV-1 transcription can correct abnormal dynamics of leukocyte infiltration in HIV-Tg, pointing to the utility of transcription inhibition in the treatment of HIV-1 associated chronic lung disease.

## 1. Introduction

Combination antiretroviral therapy (cART) significantly improves longevity of human immunodeficiency virus (HIV)-infected people, decreasing opportunistic infections. However, chronic long-term HIV infection is complicated by increased rates of age-associated chronic medical conditions such as cardiovascular, neurological, and non-infectious respiratory diseases which summarily contribute to HIV-1 morbidity and mortality [1,2,3]. Respiratory diseases are common in the HIV-1 infected patients [4]. During the cART era, prevalence of respiratory infections in HIV-infected individuals has significantly decreased, but prevalence of non-infectious lung diseases, including chronic obstructive pulmonary disease (COPD), pulmonary arterial hypertension (PAH), fibrosis and lung cancer is increasing [4,5].

COPD is a progressive disease which is characterized by irreversibly decreased airflow in association with chronic inflammation [6]. While cigarette smoking is a major risk factor of COPD, HIV-1 infection is an additional independent risk factor [7]. Patients with HIV infection, including non-smokers, are susceptible to accelerated forms of emphysema that develop significantly earlier than smoking-associated emphysema [8]. Increasing evidence suggests that lung residual macrophages orchestrate inflammation in COPD through the release of chemokines that attract neutrophils and monocytes [9]. Pathology of COPD is, in part, mediated by the improper activation of neutrophils [10,11]. Monocytes and macrophages are the major components of chronic inflammation in cART-treated patients, and both cellular and soluble myeloid associated markers have been found in association with lung disease [8,12]. Specifically, in HIV-1 patients increased levels of circulating interleukin-6(IL-6), Tumour Necrosis Factor-α (TNF-α), interleukin-1 beta (IL-1β) and C-reactive protein have been associated with chronic lung disease [13,14]. However the mechanism of increased susceptibility of HIV-positive patients for development of non-infectious respiratory diseases is not fully understood [15].

The lung is an HIV-1 reservoir in which numerous long-lived alveolar macrophages are infected with HIV-1 [16,17]. Broncho-alveolar lavages (BAL) macrophages obtained from HIV-positive patients under cART treatment contain a significant amount of integrated HIV-1 provirus that can be activated ex vivo [18,19,20]. Increased concentrations of neutrophils have been reported in the BAL obtained from HIV-1 positive patients, compared to the uninfected controls [21,22]. The lungs are constantly exposed to the different environmental insults that induce activation of lung residual macrophages that release chemokines and attract circulating neutrophils and monocytes [9]. Monocyte-derived infiltrating macrophages contribute to the resolution of early inflammation partly by digesting neutrophil apoptotic bodies [23]. The fast activation of neutrophil infiltration and fast resolution of inflammation is essential for normal lung function. Prolonged resolution of lung inflammation may significantly exacerbate lung complications. However, the dynamics of lung leukocytes infiltration and resolution of inflammation are not yet well characterized in HIV-positive individuals.

The HIV-transgenic (HIV-Tg) mouse model was developed to study HIV-associated renal disease [24,25,26,27] and lung inflammation [28]. The transgene consists of a proviral HIV-1 genome with deletion of *Gag* and *Pol* genes [25]. HIV-Tg mice express seven of the nine HIV-1 proteins under the control of viral long terminal repeat (LTR) promoter. Expression of the viral proteins is cell- and tissue-specific [29]. HIV-Tg mice express low levels of HIV transcripts in monocytes, residual macrophages and T-lymphocytes but not in the other lung cells [29,30]. This non-replicating and non-infectious HIV-Tg mouse model was recently used to study the long-term effects of viral proteins on the host [31]. The HIV-Tg mouse model is clinically relevant as it resembles a situation in cART-controlled patients, in whom there is no viral replication but there is a persistent stress due to viral protein exposure. In the pathogen-free animal facility, mice do not develop spontaneous lung complications. HIV-Tg mice are not immunocompromised and express pro-inflammatory cytokines in response to lipopolysacharide (LPS) administration [28]. LPS administration in HIV-Tg mice induces lung edema and pulmonary oxidative/nitrosative stress at 3–6 h after the administration [28]. But dynamics of lung leukocytes infiltration and resolution of inflammation in HIV-Tg mice has not yet been characterized.

Here we describe and characterize neutrophil and macrophage lung infiltration that occurs after intraperitoneal LPS administration. LPS administration induces significant lung neutrophil accumulation in both HIV-Tg and wild-type (WT) mice at 24 h post-administration, with higher accumulation levels in HIV-Tg mice. However, levels of interstitial macrophage accumulation are significantly reduced in HIV-Tg mice and accompanied by the increased levels of macrophages in the lung’s capillaries. Moreover resolution of leukocytes infiltration is significantly reduced in HIV-Tg mice. LPS administration induces significant mortality in HIV-Tg but not WT mice. In vitro trans-endothelial migration of macrophages isolated from HIV-Tg mice is impeded compared to macrophages isolated from WT mice. Treatment with a small molecule inhibitor of HIV-1 transcription 1E7-03 [32] induces trans-endothelial migration of macrophages in vitro and increases lung macrophages infiltration in HIV-Tg mice.

Taken together, our study describes the dynamics of impaired leukocytes infiltration and resolution of inflammation in the lungs of HIV-Tg mice in response to LPS administration. Inhibition of HIV-1 transcription by 1E7-03 restores lung leukocytes infiltration concurrent with the physiological response observed in WT mice. Our model might be useful for the development of novel therapeutic intervention of HIV-associated non-infectious respiratory disease and potentially might enhance the length and quality of life for patients living with HIV.

## 2. Materials and Methods

### 2.1. Experimental Design

All experiments were approved by the Howard University’s Research Institute Animal Care and Use Committee (IACUC-MED-14-09, approved October 2, 2018 until January 27, 2021). HIV-Tg breeding pairs were obtained from the Jackson Laboratory (Bar Harbor, ME, USA) and housed in a pathogen-free environment. HIV-Tg males and their wild-type (WT) littermates (5–8 weeks old, 25–30 g) were used for study. Mice received an intra-peritoneal (i.p.) injection of LPS (3 mg/kg of body weight, *Escherichia coli*, 0111:B4, Sigma-Aldrich, St. Louis, MO, USA) and were observed at various times (6, 24, 48 and 72 h). Where indicated, HIV-Tg mice also received i.p. injection of either 1E7-03 HIV transcriptional inhibitor (1 mg/kg of body weight) or 100 μL of 80% dimethyl sulfoxide (DMSO, vehicle control) 15 min after LPS challenge. Blood samples were collected through the retro-orbital venous plexus. Subsequently mice were euthanized and lungs were collected.

### 2.2. Reagents

1E7-03 was synthesized in-house using previously described synthetic scheme [32]. 1E7-03 is a tetrahydroquinoline derivative drug that inhibits HIV-1 transcription with IC_50_ = 0.9 μM and does not induce cytotoxicity at concentrations below 15 μM. It disrupts the interaction of Tat protein with host protein phosphatase 1 (PP1) [32]. Unless indicated, all chemicals were obtained from Sigma-Aldrich (St.Louis, MO, USA).

### 2.3. Immunohistochemistry

Paraffin embedded lung sections were cut at 5 μm, de-paraffinized, rehydrated, and stained as previously described [33]. Sections were labeled with primary rat anti-mouse F4/80 antibodies (Bio-Rad, Portland, ME, USA, 1:20 dilution), and rat anti-mouse neutrophil-specific antibodies (Abcam, Cambridge, MA, USA,1:200 dilution) following secondary biotinylated goat anti-rat antibodies (Dako North America Inc, Santa Clara, CA, USA). The heat-induced epitope retrieval method was used for both antibodies. The sections were incubated with streptavidin-peroxidase and developed using an 3-amino-9-ethylcarbozole (AEC) kit (both from Thermo Fisher Scientific, Waltham, MA, USA). Sections were counterstained with hematoxylin (Vector Laboratories, Inc., Burlingame, CA, USA). For the controls, the primary antibody was replaced with equivalent concentrations of the rat non-specific serum. Images were acquired by Olympus IX51 microscope with Olympus DP72 camera (both from Olympus Corporation, Waltham, MA, USA). Quantification of lung infiltrating macrophages and neutrophils was done in 10 different fields with 200× original magnification in three animals.

### 2.4. Isolation of Lung and Intra-Peritoneal Macrophages

To collect intra-peritoneal (i.p.) macrophages, mice were euthanized. About 10 mL of PBS was injected i.p. and collected 1 min later. To collect lung infiltrated macrophages, lungs were cut into small pieces (1–2 mm), pieces were pushed through the syringe needle (18G), suspended in 5 mL of PBS with 0.5 mM ethylendiaminetetraacetic acid (EDTA) incubated for 30 min at +4 °C, and then filtered through 100 μm cell strainer. Both lung-derived and i.p. infiltrating cells were concentrated by centrifugation at 1500× *g* and suspended in 1ml of DMEM cell culture medium (Thermo Fisher Scientific, Waltham, MA, USA) with 10% DMSO. Cells were stored at −80 °C until use.

### 2.5. Generation of Mouse Lung Endothelial Cell Line

Lung tissue was collected from WT mouse and lung vessels were isolated. Small vessels were cut into the 1 mm pieces and incubated with Collagenase IV (1mg/mL, Sigma Aldrich) for 1 h at 37 °C. Cells were purified through the 70 µm cell strainer (Sigma-Aldrich, St. Louis, MO, USA) and incubated in the endothelial CSC complete media (Cell Systems Corporation, Kirkland, WA, USA) for 7 days. Endothelial cell clones were infected with Ad.SV-40 virus (2 × 10^9^ particles/mL). Three days after infection cells were transferred into DMEM media supplemented with 10% FBS and antibiotics (all from Thermo Fisher Scientific, Waltham, MA, USA) and maintained for 4–6 weeks. Emerging clones were maintained separately. No growing clones were found in the control cells without Ad.SV-40 infection. Transformed cells were positive for endothelial marker vWF (Supplemental Figure A1A, Appendix A).

### 2.6. Trans-Endothelial Migration Assay

Trans-endothelial migration of mouse macrophages was assessed by modified Boyden chamber assay using peritoneal macrophages. Mice were injected i.p. with LPS (0.3 μg/mg of body weight), and peritoneal lavage was collected 48 h after injection. Cells were washed with PBS and quantified. At least four mice were used for each experiment. Briefly, 1 × 10^5^ endothelial cells were placed on the polycarbonate supports in tissue culture inserts (Transwell-COL, Thermo Fisher Scientific, Waltham, MA, USA) and incubated until formation of a confluent monolayer. Formation of monolayer was monitored by measurement of trans-endothelial resistance using a voltammeter (EVOM) with an ENDOHM chamber (both from World Precision Instruments, Sarasota, FL, USA). Mouse lung endothelial cells formed confluent monolayer with resistance 67.4 ± 14.7 om/cm^2^ (*N* = 10). About 5 × 10^4^ mouse peritoneal macrophages were added on the top of endothelial cell monolayer and incubated for 72 h. To stimulate macrophage migration into the low chamber, monocyte chemoattractant protein 1 (10 ng/mL MCP1, R&D Systems, Minneapolis, MN, USA) was added to the low chamber. Macrophages were collected from the low chamber and quantified using Trypan blue assay (Thermo Fisher Scientific, Waltham, MA, USA).

Total RNA was isolated from collected macrophages, and quantitative real-time polymerase chain reaction (RT-PCR) was performed to analyze HIV-1 *env* gene expression. To inhibit HIV-1 transcription, 1E7-03 inhibitor (1 µM) was added to the macrophages in the upper chamber.

### 2.7. Real-Time Polymerase Chain Reaction (RT-PCR)

Total RNA was extracted from peritoneal macrophages using TRIzol (Thermo Fisher Scientific, Waltham, MA, USA) and was column-purified using an RNA isolation kit (Ambion Thermo-Fisher Scientific). First-strand cDNA was prepared from total RNA using the SuperScript II First Strand Synthesis kit (Thermo Fisher Scientific). RT-PCR was performed in triplicates using Platinum Taq DNA polymerase (Thermo Fisher Scientific). The HIV-1 *env* was amplified with primers: forward 5′-TGTGTAAAATTAACCCCACTCTG, reverse 5′-ACAACTTGTCAACTTATAGCTGGT-3′; and HIV-1 *tat* with primers: forward 5′- ATGGAGCCAGTAGATCCTAGAC-3′, reverse 5′-CTAATCGAATCGATCTGTCTCTGC-3′. Mouse *Gapdh* was amplified with primers: forward 5′- GCCAAGGTCATCCATGACAAC-3′, reverse 5′-CTTACTCCTTGGAGGCCATGT-3′, and was used for normalization.

### 2.8. Lung Injury Scores

Lung injury scores were determined by the lung injury scoring system (American Thoracic Society, 2010). Hematoxylin and eosin-stained lung sections were evaluated under 400× magnification using Olympus IX51 microscope with Olympus DP 72 camera. Evaluation of parameters was done in fifteen random fields in three animals, and lung injury scores were calculated as described in the Table 1.

Lung injury scores were calculated using equation:Score = (20i + 14ii + 7iii + 7iv + 2v)/100

### 2.9. Bleeding Score

Hematoxylin and eosin stained lung sections were evaluated with 200× magnification using Olympus IX51 microscope with Olympus DP 72 camera. Bleeding scores were set as: 0—no bleeding; 1—bleeding in 1 alveoli; 2—bleeding in 2–3 alveoli; 3—bleeding in 4–5 alveoli; and 4—bleeding in more than 5 alveoli. Bleeding scores were evaluated in fifteen random fields for three animals per group.

### 2.10. Flow Cytometry

For immunostaining, 5 × 10^5^ cells were incubated with Fc Block-2.4G2 (BD Biosciences, Franklin Lakes, NJ, USA) antibody to block Fcγ III/II receptors in 10% goat serum for 10 min before the addition of fluorochrome-conjugated anti-mouse antibodies: anti mouse macrophage F4/80-FITC and anti-mouse neutrophil Ly-6-PE (all from BD Biosciences). Cells were incubated with antibodies for 30 min on ice, washed with 1 mL of PBS, suspended in 0.5 mL of PBS and used for flow cytometry on a FACSVerse (BDBiosiences).

Flow cytometry analysis was conducted in duplicate. Quantification of acquired flow cytometry data for lung macrophages and neutrophils was performed in three mice for each treatment at each time point.

### 2.11. BioPlex Cytokine Analysis

Plasma samples were collected from HIV-Tg and WT animals at 24 h post-LPS administration and kept frozen until use. Bio-Plex Pro Mouse group 1 Th1 panel L60-00004C6 kit was obtained from Bio-Rad, Portland, ME, USA. Samples from three mice were used for each group. The assay was performed using the Bio-Plex suspension array system according to the manufacturer’s instructions (Bio-Rad). In brief, the appropriate cytokine standards and samples were added to a 96-well flat plate. The samples were incubated at room temperature for 30 min with antibodies chemically attached to fluorescent-labeled magnetic micro beads. After three washes on Bio-Plex ProII wash station, premixed detection antibodies were added to each well and incubated for 30 min. Following three washes, premixed streptavidin-phycoerythrin was added to each well and incubated for 10 min followed by three more washes. Then beads were re-suspended with 125 µL of assay buffer and the reaction mixture was quantified using the Bio-Plex protein array reader. Data were automatically processed and analyzed by Bio-Plex Manager Software 6.0 using the standard curve produced from recombinant cytokine standard.

### 2.12. Mouse VEGF Enzyme-Linked Immunosorbent Assay (ELISA)

Plasma VEGF levels were determined using mouse VEGF Quantikine enzyme-linked immunosorbent assay (ELISA) kit (R&D Systems, Minneapolis, MN, USA) according to manufacturer’s instruction. Samples from three mice were used for each group.

### 2.13. Statistical Analysis

All data are presented as mean and standard deviation. Statistical analysis was performed using GraphPad Prizm 6 software (Graph Pad Software, San Diego, CA, USA). Differences between the two groups were compared using the Student’s *t*-test. When more than two groups were compared, we used one-way analysis of variance (ANOVA) followed by multiple pair-wise comparisons using the Student-Newman-Keuls test. Survival curves were analyzed using the Kaplan–Meier survival analysis. *p* values < 0.05 were considered significant.

## 3. Results

### 3.1. Increased Neutrophil Infiltration and Decreased Macrophage Infiltration in Lungs of Human Immunodeficiency Virus 1 Transgenic (HIV-Tg) Mice after LPS Administration

Intraperitoneal injection of LPS induced profound inflammation in both HIV-Tg and WT mice at 6 h post injection (Figure 1). Lung injury was characterized by pulmonary edema (Figure 1A), hemorrhage (Figure 1B,D, indicated by red blood cells in alveoli and black arrowheads) and interstitial neutrophils infiltration (Figure 1C,E, indicated by yellow arrowheads).

Pulmonary edema is a major complication of LPS-induced acute lung injury [34]. To characterize the edema, wet-to-dry ratios of lung tissue were determined (Figure 1A). The lung wet-to-dry ratios for non-injected HIV-Tg mice were previously shown to be higher than for non-injected WT mice, indicating that the lungs of HIV-Tg mice have edema and predisposition to pulmonary complications as previously reported [28]. To confirm the edema, wet-to-dry ratios of lung tissue were assessed (Figure 1A). Lungs of HIV-Tg mice injected with PBS demonstrated higher wet-to-dry ratios than lungs of WT mice injected with PBS (Figure 1A, WT-PBS versus HIV-PBS, *p* = 0.05). LPS injections increased wet-to-dry ratios in WT as well as in HIV-Tg mice (Figure 1A). This increase was statistically significant for WT mice (Figure 1A, WT-PBS versus WT-LPS, *p* = 0.03) but not for HIV-Tg mice because of pre-existing edema (Figure 1A, HIV-PBS versus HIV-LPS, *p* = 0.21). Also, no significant differences in the wet-to-dry ratios between WT and HIV-Tg mice injected with LPS were found (Figure 1A, WT-LPS versus HIV-LPS, *p* = 0.23). LPS injection induced bleeding in both HIV-Tg and WT mice, but no differences in bleeding scores were found (Figure 1B,D, WT-PBS versus WT-LPS, *p* = 1.3 × 10^−9^; HIV-Tg-PBS versus HIV-Tg-LPS, *p* = 8.8 × 10^−10^; and WT-LPS versus HIV-LPS, *p* = 0.58). LPS administration induced significantly higher lung injury scores compared to PBS injection in both WT and HIV-Tg mice (Figure 1C, WT-PBS versus WT-LPS, *p* = 0.002; and HIV-PBS versus HIV-LPS, *p* = 2.3 × 10^−5^). LPS injection induced lung injury in both HIV-Tg and WT mice, but no differences were found in lung injury scores between WT and HIV-Tg mice (Figure 1C, WT-LPS versus HIV-LPS, *p* = 0.83). The major finding was the observation of interstitial neutrophils infiltration (Figure 1D,E, yellow arrowheads) and capillary microthrombi formation (Figure 1E, black arrow). Taken together, our results demonstrated similar levels of lung injury in WT and HIV-Tg mice at 6 h post intraperitoneal LPS administration.

In contrast, at 24 h post LPS administration HIV-Tg mice demonstrated significantly higher levels of lung injury (Figure 2A,B, WT versus HIV, *p* = 0.0001). Patchy neutrophilic infiltrates and thickening alveolar walls with interstitial neutrophils were more prominent in HIV-Tg mice compared to WT mice (Figure 2A, arrowheads). Immunostaining of neutrophils demonstrated over two-fold increased accumulation in LPS-injected HIV-Tg animals compared to LPS-injected WT mice (Figure 2C,D, red color, WT versus HIV-Tg PNMC, *p* = 0.0051). Levels of lung neutrophils and macrophages were similar in control groups of WT and HIV-Tg mice injected with PBS (Supplemental Figure A2A,B, PBS injected WT versus for HIV-Tg PNMC, *p* = 0.5476; for Mϕ, *p* = 0.6018). Injection of LPS stimulated significant interstitial lung macrophage infiltration in WT mice (Figure 2E,F, red staining). In contrast, in LPS-injected HIV-Tg mice there were much less infiltrated interstitial macrophages compared to WT mice (Figure 2E,F, WT versus HIV-Tg, red staining, *p* = 0.0057).

Interestingly, more F4/80 positive macrophages were observed in capillaries of HIV-Tg mice injected with LPS compared to LPS-injected WT mice (Figure 2G,H, red staining, WT versus HIV, *p* = 0.0001) suggesting a possible defect in the macrophage endothelial transmigration in HIV-Tg mice.

To quantify lung infiltrating leukocytes, we isolated them as described in Materials and Methods and analyzed by FACS (Figure 3 and Supplemental Figure A3). The gate strategy for neutrophils and macrophages isolated from HIV-Tg and WT mice is shown in Supplemental Figure A3. In LPS-injected HIV-Tg mice, levels of lung infiltrated macrophages were reduced (Figure 3A, F4/80-FITC blue color—WT, red color—HIV-Tg, and quantification in Figure 3B) but levels of lung infiltrated neutrophils were increased (Figure 3C, Ly-6G-PE blue color—WT, red color—HIV-Tg, and quantification in Figure 3D) at 24 h post injection. Correspondingly, the neutrophil/macrophage ratios were significantly higher in HIV-Tg mice compared to WT mice (Figure 3E, *p* = 0.018).

Taken together, levels of lung infiltrating neutrophils and macrophages were significantly altered in HIV-Tg mice at 24 h after LPS administration compared to WT mice.

### 3.2. LPS Injections Induced Higher Levels of Inflammatory Cytokines in HIV-Tg Mice

To characterize the LPS-associated inflammation, expression of pro-inflammatory cytokines was examined in the serum at 24 h post injection using a Bio-Rad suspension array cytokine panel (Figure 4). IL-1β and IL-2 levels were similar in WT mice injected either with PBS or with LPS (Figs.4A and 4B). In contrast, levels of IL-1 (Figure 4A, HIV-Tg-PBS versus HIV-Tg-LPS, *p* = 0.045) and IL-2 (Figure 4B, HIV-Tg-PBS versus HIV-Tg-LPS, *p* = 0.046) were significantly higher in HIV-Tg mice injected with LPS. LPS administration also significantly increased IL-6 levels (Figure 4C, WT-PBC versus WT-LPS, *p* = 0.015; and HIV-Tg-PBS versus HIV-Tg-LPS, *p* = 0.018) and TNF-α levels (Figure 4D, WT-PBS versus WT-LPS, *p* = 0.008; and HIV-Tg-PBS versus HIV-Tg-LPS, *p* = 0.015) compared to PBS injections in both HIV-Tg and WT mice.

The levels of IL-6 were significantly higher in HIV-Tg mice injected with LPS compared to LPS-injected WT mice (Figure 4C, WT-LPS versus HIV-Tg-LPS, *p* = 0.023). We also tested serum levels of vascular endothelial growth factor (VEGF), which is a major factor of endothelial permeability [35]. In airways, VEGF induces vascular leakage and airway edema, and it enhances chemotaxis for monocytes in COPD [36]. Serum VEGF levels were similar in WT and HIV-Tg mice injected with PBS (Figure 4E, WT-PBS versus HIV-Tg-PBS, *p* = 0.16). LPS administration significantly increased serum VEGF levels in both WT and HIV-Tg mice (Figure 4E, WT-PBS versus WT-LPS, *p* = 0.05; and HIV-Tg-PBS versus HIV-Tg-LPS, *p* = 0.017). Overall serum levels of IL-1β, IL-2, TNF-α and VEGF were slightly higher, and IL-6- significantly higher in LPS-injected HIV-Tg mice compared to LPS-injected WT mice. These results verified that HIV-Tg mice were not immunocompromised and exhibited higher inflammatory response to LPS than WT mice in accord with the previous study [28]. Thus, reduced levels of lung macrophages infiltration in HIV-Tg mice were not associated with reduced levels of inflammatory cytokines.

### 3.3. Resolution of Lung Leukocytes’ Infiltration after LPS Administration Is Slower in HIV-Tg Mice

Sensitivity to LPS-mediated injury varies significantly among mouse strains [37]. Treatment of FVB mice with low doses of LPS (3 mg/kg of body weight) has been shown to induce inflammation and significant lung injury, but it is quickly resolved and all treated mice recover from the insult [38]. Moreover, the recovery time from LPS-induced injury correlates with the time course resolution of lung leukocytes infiltration. Thus, we next evaluated the resolution time of lung neutrophil and macrophage infiltration at 24–72 h post LPS administration using FACS (Figure 5A,B). The peak of leukocyte infiltration was detected at 48 h post LPS administration in both WT and HIV-Tg mice.

The cell number of neutrophils significantly increased between 24 and 48 h in both mouse strains (Figure 5A, WT 24 h versus 48 h, *p* = 0.011; and HIV-Tg 24 h versus 48 h, *p* = 0.046). Macrophage lung infiltration was also significant in both strains, but it had different kinetics for WT and HIV-Tg mice (Figure 5B). While for WT mice, macrophages infiltration peaked at 24 h and was significantly reduced at 72 h, in HIV-Tg mice, the infiltration was delayed, with the peak at 48 h and little reduction at 72 h (Figure 5B). Therefore, both neutrophil and macrophage infiltrations were significantly reduced at 72 h in WT mice (Figure 5A, compare WT at 48 h and 72 h, *p* = 0.015; and Figure 5B, compare WT at 48 h and 72 h, *p* = 0.016). In contrast, in HIV-Tg mice neutrophil levels were significantly reduced at 72 h after LPS administration (Figure 5A, compare HIV-Tg at 48 h and 72 h, *p* = 0.03), but macrophage levels remained similar at 48 h and 72 h (Figure 5B, compare HIV-Tg at 48 h and 72 h, *p* = 0.42). Neutrophil levels were persistently elevated at 24 h and 72 h after LPS injections in HIV-Tg mice compared with WT mice (Figure 5A, WT versus HIV-Tg at 24 h, *p* = 0.012; and 72 h, *p* = 0.07). Interestingly, macrophage levels were reduced at 24 h and 48 h and increased at 72 h in HIV-Tg mice compared to WT mice (Figure 5B, WT versus HIV-Tg). Lung immunostaining also demonstrated increased levels of neutrophil and macrophage accumulation in HIV-Tg mice at 72 h (Figure 5C, PNMC and Figure 5E, Mϕ, red staining) but quantification of lung leukocytes showed no statistical significance, possibly due to the patchy distribution of lung leukocytes (Figure 5D, PNMC, *p* = 0.1470; and Figure 5F, Mϕ, *p*=0.1204).

In addition, a marked difference was observed in the mortality of WT and HIV-Tg mice injected with LPS. All WT mice survived LPS injection and returned to the normal physiological activity at 48 h after LPS administration (Figure 5G, black line). In contrast, about 45% of HIV-Tg animals injected with LPS (4 out of 9) died at day 3 post-injection (Figure 5G, red line). No HIV-Tg animals injected with PBS died (*N* = 10, not shown). Previously we demonstrated that small molecule 1E7-03 inhibited HIV-1 transcription in T cells and macrophages in vitro [32] and reduced HIV-1 replication in humanized mouse model [39]. Injection of 1E7-03 (1 mg/kg of body weight) to HIV-Tg mice prevented HIV-Tg mice mortality as all HIV-Tg mice injected with 1E7-03 in combination with LPS (*N* = 6) survived for 5 days (Figure 5G, blue line).

These results are in line with our previous observation that FVB WT mice quickly recover from LPS-induced inflammation at 3–5 days post injection [38]. In contrast, LPS administration in HIV-Tg mice induced significant mortality, which was associated with lung injury and accumulation of neutrophils and macrophages.

### 3.4. Trans-Endothelial Migration of Macrophages Isolated from HIV-Tg Mice Are Reduced In Vitro

Because we found significant accumulation of macrophages in the lung capillaries of HIV-Tg mice (Figure 2G,H), we considered a possibility of migration defect in HIV-Tg macrophages. To test trans-endothelial migration of HIV-Tg mouse macrophages in vitro we used modified Boyden chamber assay. Altered migration of HIV-expressing macrophages on different substrates was demonstrated previously [40], but the effect of HIV expression on trans-endothelial migration was not evaluated.

To study trans-endothelial migration of macrophages, we generated murine lung endothelial cell line as described in Materials and Methods. Endothelial cells were seeded on membrane inserts to form a monolayer. Peritoneal macrophages were isolated from HIV-Tg and WT mice and added to the endothelial cell monolayer in the insert as described in Materials and Methods. Expression of HIV-1 *env* and *tat* genes was detected in isolated macrophages by RT-PCR (Figure 6A).

In order to stimulate macrophage migration to the low chamber mouse macrophage-monocyte chemoattractant protein 1 (MCP1-CCL2, 8 ng/mL) was added. Macrophages were collected from the low chamber after 72 h incubation and quantified using Trypan blue assay (Figure 6B). Both trans-endothelial migration (Figure 6B) and chemotaxis (Figure 6C, migration through the insert’s membrane without endothelial cells) were tested. The number of HIV-Tg mouse macrophages that migrated to the low chamber through endothelial monolayer was significantly lower compared to WT mouse macrophages (Figure 6B, WT-vehicle versus HIV-Tg-vehicle, *p* = 6 × 10^−3^). In contrast, similar amounts of HIV-Tg and WT mouse macrophages migrated through empty membrane (Figure 6C, WT-vehicle versus HIV-Tg-vehicle, *p* = 0.96). These results further support the hypothesis of abnormal trans-endothelial migration of HIV-Tg mouse macrophages.

### 3.5. Administration of HIV-1 Transcription Inhibitor 1E7-03 Improves Trans-Endothelial Macrophages Migration In Vitro and the Levels of Lung Leukocytes Infiltration in HIV-Tg Mice

HIV-Tg mice do not produce viral particles, and therefore cART is not effective in HIV-Tg mice. However HIV-1 gene expression in the peritoneal macrophages indicated an active viral transcription in this model (Figure 6A). To determine whether HIV-1 gene expression impacts macrophages trans-endothelial migration, we treated macrophages with HIV-1 transcriptional inhibitor 1E7-03 [32]. 1E7-03 (1µM) was added to the macrophages in the upper chamber in Boyden chamber assay and cells were incubated for 72 h. Dimethyl sulfoxide (DMSO) (10%) treatment was used as a negative control (vehicle). Treatment with 1E7-03 significantly increased trans-endothelial migration of HIV-Tg macrophages (Figure 6B, HIV-Tg-vehicle versus HIV-Tg-1E7-03, *p* = 0.026) without affecting chemotaxis (Figure 6B, chemotaxis, HIV-Tg-vehicle versus HIV-Tg-1E7-03, *p* = 0.55). 1E7-03 treatment did not exhibit cell toxicity and did not induce macrophages or endothelial cell death (Supplemental Figure A1B, endothelial cell number vehicle versus 1E7-03 treatment, *p*=0.12; Figure A1C viability of mouse macrophages vehicle versus 1E7-03 treatment, *p* = 0.77; Appendix A).

As we demonstrated, lung injury and inflammation were significantly increased in HIV-Tg mice compared to WT mice at 24 h after LPS administration (Figure 2 and Figure 4). Injection of 1E7-03 (1 mg/kg of body weight) to HIV-Tg mice reduced serum levels of IL-6 to levels observed in WT mice (Figure 6D, HIV-Tg-LPS vs. HIV-Tg-LPS-1E7-03, *p* = 0.028). Moreover 1E7-03 administration completely prevented hemorrhage in LPS injected mice (Figure 6E, WT-LPS vs. WT-LPS-1E7-03, *p* = 5.6x10^-4^; HIV-Tg-LPS vs. HIV-Tg-LPS-1E7-03, *p* = 4.3 × 10^-7^) and significantly reduced lung injury scores in HIV-Tg mice (Figure 6F, HIV-LPS versus HIV-LPS-1E7-03, *p* = 0.03).

Administration of 1E7-03 into WT mice did not change percent of lung macrophages (Figure 7A, F4/80 staining, WT, blue color—vehicle, red color—1E7-03; Figure 7B, quantification of FACS results) and neutrophils (Figure 7C, Ly-6-PE staining, WT blue color—vehicle, red color—1E7-03; Figure 7D, quantification) at 48 h after administration. In contrast, administration of 1E7-03 into HIV-Tg mice significantly increased macrophage levels (Figure 7A, HIV-Tg, blue color—vehicle, red color—1E7-03, *p* = 0.006; Figure 7B, quantification) and reduced neutrophil levels (Figure 7C, HIV-Tg, blue color—vehicle, red color—1E7-03, *p* = 0.006; Figure 7D, quantification).

Lung immunostaining also demonstrated that 1E7-03 injections significantly increased levels of lung interstitial macrophages (Figure 7E, compare HIV-LPS and HIV-LPS-1E7-03) and reduced levels of neutrophil lung infiltration (Figure 7F, compare HIV-LPS and HIV-LPS-1E7-03). All HIV-Tg mice injected with 1E7-03 in combination with LPS survived (Figure 5G, blue line).

Taken together, administration of HIV-1 transcription inhibitor 1E7-03 improved trans-endothelial migration of HIV-Tg macrophages in vitro and the levels of lung neutrophil and macrophage infiltration in HIV-Tg mice concurrent with the physiological response observed in WT mice.

## 4. Discussion

HIV-infected individuals on cART therapy have higher prevalence of non-infectious lung disease than HIV negative individuals [41]. Repeated immune system activation associated with recurrent increase of circulating LPS due to systemic translocation of microbial products across the damaged gut mucosal barrier might be a risk factor for development of chronic inflammation and immune system activation in the peripheral organs. Leukocyte recruitment from circulation into the site of inflammation is a fundamental process in the inflammatory response. Timely and well-orchestrated recruitment of leukocytes and resolution of inflammation is essential for prevention of organ injury. Monocytes and macrophages are permissive for HIV-1 infection. HIV-1 infection significantly affects macrophages functions, altering cytokines expression, phagocytosis and migration [40,42,43]. As HIV-1 does not infect neutrophils, it may impede their function due to alterations of systemic immune response.

In this study, the intra-peritoneal administration of a non-lethal dose of LPS was used to mimic microbial products translocation without active bacterial infection. The dose was selected based on our previous study in FVB mice [38]. In HIV-Tg mice, HIV-1 proteins are expressed at low levels in the monocytes and lymphocytes without active virus replication, mimicking cART-controlled or latent HIV-1 infection in humans. LPS does not activate HIV-1 gene expression but induces inflammation stimulating neutrophils and macrophages infiltration in organs [44,45]. Therefore, this model provides an opportunity to study the role of macrophages expressing HIV-1 transcripts in the development of lung injury in the absence of active viral and bacterial infections.

Intraperitoneal injection of LPS induced profound inflammation in both HIV-Tg and WT mice as early as 6 h post injection, but there was not any difference in lung injury between WT and HIV-Tg mice at that early time point. During the course of immune activation and resolution of inflammation, LPS-induced lung injury in HIV-Tg mice was significantly exacerbated. Here we demonstrated altered dynamics of lung leukocytes infiltration and its resolution in HIV-Tg mice.

LPS administration is also known to induce secretion of pro-inflammatory cytokines and chemokines by resident alveolar macrophage which leads to the recruitment of leukocytes from circulation [46]. Neutrophils are the first responding cells that play a critical initial role in controlling lung infections by phagocytosing the microorganisms and releasing mediators of delayed alveolar monocytes recruitment [47,48]. Our flow cytometry results demonstrated that LPS administration significantly increased neutrophil levels in both WT and HIV-Tg mice at 24 h after injection. Neutrophil levels were significantly higher in HIV-Tg mice. In contrast, lung macrophage levels were significantly lower in HIV-Tg mice.

The infiltration of leukocytes into organs is stimulated by inflammation. Because levels of circulating IL-1β, IL-6 and TNF-α are increased in COPD [14] we tested these cytokine levels in HIV-Tg and WT mice after LPS administration. In agreement with the previous study [28] we did not observe significant differences in the levels of circulating pro-inflammatory cytokines (IL-1β, IL-2, and TNF-α) except for IL-6 at 24 h post LPS administration, suggesting overall similar levels of inflammation in both HIV-Tg and WT mice. Thus, the HIV-Tg mouse, which carries non-replicating and non-infectious HIV-1 transgene, is not immunocompromised as was verified by cytokines expression analysis. Only IL-6 levels were significantly increased in HIV-Tg mice compared to WT mice. This increase might be associated with higher IL-6 production by organ residual macrophages that express HIV genes in HIV-Tg mice. Enhanced production of IL-6 in response to LPS was previously demonstrated in human macrophages infected ex vivo with HIV-1 [49]. In addition to the alveolar macrophages, LPS-also activates endothelium which may represent an another source of inflammation in the mouse model [50]. Because endothelial cells do not express HIV genes in HIV-Tg mice [25], it is unlikely that these cells produce significantly higher levels of cytokines and chemokines upon LPS stimulation in HIV-Tg mice than in WT mice.

The flow cytometry is widely used for the analysis of lung leukocytes but the limitation of this approach is the inability to distinguish cells that infiltrated lung parenchyma from the cells that remained in the pulmonary blood vessels. Thus, to achieve a more comprehensive, mechanistic understanding of lung leukocytes infiltration, we performed immunostaining of lung neutrophils and macrophages. These experiments pointed not only to the reduced levels of lung infiltrating macrophage in HIV-Tg mice, but also indicated significant abnormalities in macrophages distribution. Significantly lower number of macrophage was found to infiltrate alveoli and interstitial spaces of HIV-Tg animals compared to WT mice. Moreover, macrophages were accumulated in the lung capillaries attaching to the endothelium in HIV-Tg animals. Thus, migration of macrophages in HIV-Tg mice was significantly different from macrophage migration in WT mice. The passage of leukocytes across the blood vessel wall is a fundamental event in the inflammatory response. LPS administration induces leukocytes attachment to the endothelial cells, diapedesis and infiltration into the organ matrix. These events are accompanied by the increased permeability of endothelial cells. We observed increased endothelial cell permeability after LPS administration in both WT and HIV-Tg mice as reflected by the similar levels of VEGF and wet/dry lung ratios. Thus, it was unlikely that reduced endothelial permeability played a major role in the macrophage migration discrepancy in HIV-Tg mice. Alternatively, expression of HIV genes in macrophages might mediate their migration. HIV-infection affects macrophages migration, reducing amoeboid migration mode on the matrix surface and enhancing mesenchymal migration in 3D matrix in vitro [40]. Here we characterized trans-endothelial migration of macrophages in vitro, comparing macrophages isolated from HIV-Tg and WT mice. Macrophages were isolated from the peritoneal cavity of WT and HIV-Tg mice, and expression of HIV genes (*env* and *tat*) was demonstrated in HIV-Tg macrophages. Interestingly, migration of macrophages isolated from HIV-Tg mice through the layer of mouse lung endothelial cells was significantly reduced compared to WT macrophages. In contrast, migration of both HIV-Tg and WT macrophages through a membrane that lacked endothelial cells (chemotaxis) was similar. Therefore, reduced trans-endothelial migration of macrophages is likely to be a reason for reduction of lung macrophages in HIV-Tg mice. As we used endothelial cells isolated from the lung of the WT mouse, we can rule out the possibility of migration deficiency being associated with differential expression levels of endothelial cell-adhesion molecules, leukocyte activators or chemokines. Thus, we assumed that reduction of HIV-Tg macrophages migration was associated with HIV gene expression. To confirm that HIV-1 gene expression was critical for reduction of macrophages migration, we utilized small molecule inhibitor of HIV-transcription, 1E7-03, which we previously showed to inhibit HIV-1 transcription in vitro [32] and in vivo [39]. Treatment with 1E7-03 significantly improved macrophage trans-endothelial migration in vitro without effecting chemotaxis. Moreover, administration of 1E7-03 in HIV-Tg mice remarkably restored lung macrophage infiltration to the levels observed in WT mice and prevented mice mortality observed with LPS administration. 1E7-03 significantly reduced lung neutrophil accumulation and improved lung injury scores. Taken together, these observations point to the critical role of HIV-1 gene expression in the impairment of macrophage lung migration even in the absence of active virus production in HIV-Tg mice.

In addition to the altered macrophages migration at the early time point (24 h), our data demonstrated a delayed resolution of inflammation in HIV-Tg mice. We found that neutrophil and macrophage lung infiltrations reached their peaks at 48 h post LPS administration in WT mice. Subsequently neutrophil infiltration was almost completely resolved at 72 h. In contrast, levels of neutrophil infiltration in HIV-Tg mice were significantly higher at early (24 h) and late (72 h) time points. Macrophage levels in HIV-Tg mice were relatively low at the early time point (24 h) and progressively increased at 48 h and 72 h. The timely resolution of neutrophil inflammation is necessary for the prevention of tissue injury, and inadequate resolution and failure to return tissue to homeostasis results in neutrophil-mediated destruction and chronic inflammation [23]. Macrophages contribute to the resolution of early inflammation by digesting neutrophil apoptotic bodies. Thus, the increase of lung neutrophil accumulation in HIV-Tg mice was likely due to the inability of the monocyte-derived macrophages to reach the neutrophil-occupied spaces. Consequently, reduced resolution of neutrophil infiltration and exacerbation of neutrophil-mediated organ injury might play a critical role in the observed higher mortality of HIV-Tg mice. Previously we demonstrated that the FVB mouse strain, used for the generation of HIV-Tg mice, was resistant to LPS-induced injury. The lung inflammation in FVB mice was resolved within 3-5 days after LPS administration [38]. In contrast, lung neutrophil and macrophage levels in HIV-Tg mice were increased at 48-72 h and this was accompanied by mortality of 45% of HIV-Tg mice. LPS-induced mortality in HIV-Tg mice might be associated with a “cytokine storm” accompanying a higher level of immune system activation and delayed resolution of neutrophil infiltration. Inhibition of HIV-1 transcription prevented LPS-associated mortality in HIV-Tg mice and lung injury suggesting that HIV-1 transcription can be targeted therapeutically to prevent HIV-1 associated lung disease. LPS administration in HIV-Tg mice induced pulmonary oxidative and nitrosative stress additionally to immune dysfunction [28]. Future experiments will assess the role of redox stress in the mediation of pulmonary complication in HIV-Tg mice.

Our study only focused on the abnormal macrophage infiltration and resolution of inflammation in the lungs of HIV-Tg mice. Thus, it would be interesting to evaluate the acute immune response and resolution of inflammation in other organs of HIV-Tg mice.

In contrast to the HIV-infected patients where only a fraction of macrophages is infected with HIV-1, the HIV-1 transgene is present in all macrophages of HIV-Tg mice. Thus, HIV-1 genes are likely to be expressed in most macrophages in HIV-Tg mice leading to profound pathology. In HIV-1 patients, for whom only a fraction of macrophages are infected, it is not likely that lung neutrophils’ accumulation will lead to acute mortality. However, impaired macrophage migration might be a contributing factor to the persistent lung inflammation in HIV-1 patients leading to COPD development and exacerbation. Because cART therapy does not affect HIV transcription, development of transcriptional inhibitors such as 1E7-03 for clinical use might be helpful for treatment of chronic lung inflammation in HIV-1 positive individuals presenting with chronic lung disease.

## 5. Conclusions

In conclusion, HIV-1 expression in macrophages in the absence of productive viral infection affects the dynamics of macrophage infiltration and neutrophil resolution in the lungs of HIV-Tg mice. Thus, the phenomenon has to be considered in future therapeutic approaches and development of novel drugs to treat chronic complications of HIV-1 infection.

## Figures and Tables

**Figure 1 viruses-12-00204-f001:**
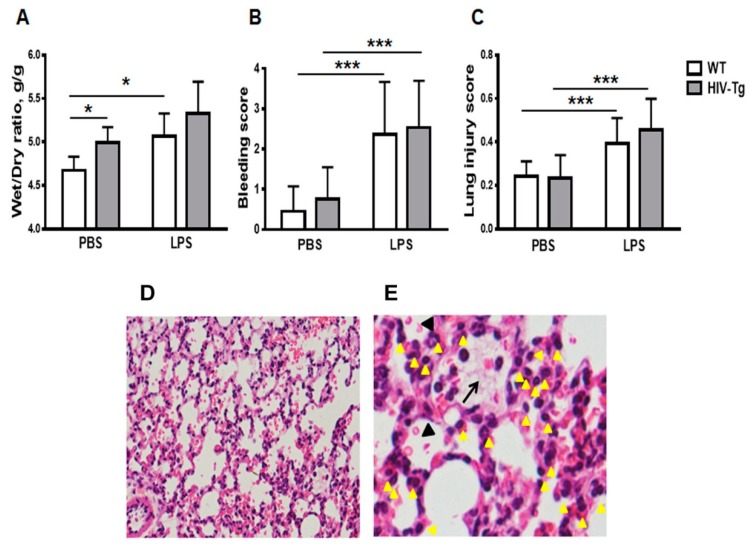
Intraperitoneal lipopolysaccharides (LPS) injections induce similar lung injury in WT and HIV-Tg mice at 6 h post administration. (**A**). Wet-to-dry ratios of lung tissue, (**B**) lung hemorrhage score and (**C**) lung injury score in wild-type (WT) and human immunodeficiency virus 1 transgenic (HIV-Tg) mice at 6 h after administration of either PBS or LPS. Mean and standard error are shown. Asterisks indicate: * *p* < 0.05 and *** *p* < 0.001. (**D**,**E**) Representative images of lung sections from HIV-Tg mouse injected with LPS and stained with hematoxylin and eosin. Original magnifications 100× (**D**) and 400× (**E**). Interstitial infiltrated neutrophils are indicated by yellow arrowheads; capillary microtrombi formation is indicated by black arrow; and hemorrhage is indicated by black arrowhead.

**Figure 2 viruses-12-00204-f002:**
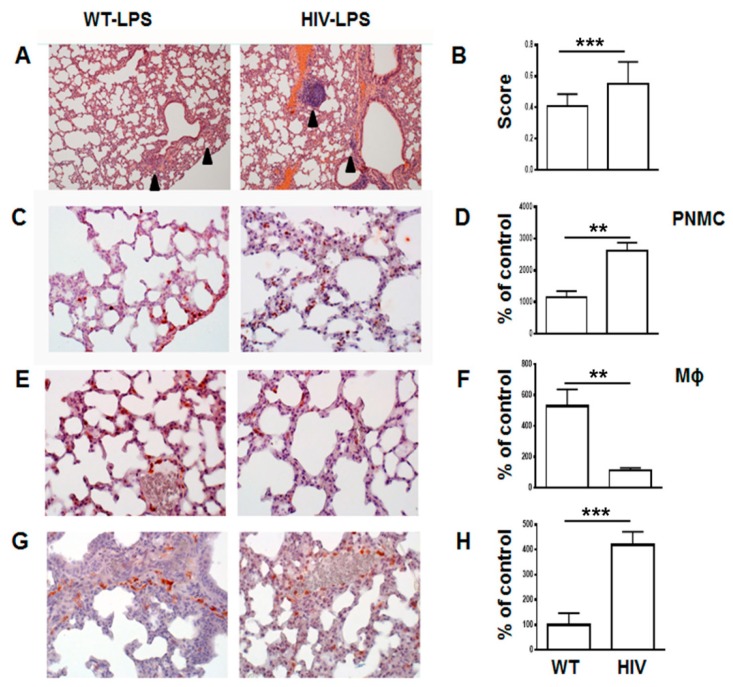
Significantly increased lung injury in HIV-Tg mice at 24 h post LPS administration. (**A**) Lung histology at 24 h post LPS administration. Arrowheads indicate immune cells infiltration. Hematoxylin and eosin staining. Original magnification 100×. (**B**) Lung injury score. Mean and standard error are shown. (**C**) Immunostaining of lung neutrophils (red color). Original magnification 400×. (**D**) Quantification of lung neutrophils infiltration (PNMC). Results are shown as percent of WT-PBS level. Mean and standard error are shown. (**E**) Immunostaining of lung macrophages (red color). Original magnification 400×. (**F**) Quantification of lung macrophages infiltration (Mϕ). Results are shown as percent of WT-PBS level. Mean and standard error are shown. (**G**) Immunostaining of macrophages accumulating in lung capillaries (red color). Original magnification 400×. (**H**) Quantification of lung capillary macrophages (Capillary Mϕ). Results are shown as percent of WT-PBS level. Mean and standard error are shown. Hematoxylin is used for counterstaining (purple nuclear staining). Asterisks indicate: ** *p* < 0.01 and *** *p* < 0.005.

**Figure 3 viruses-12-00204-f003:**
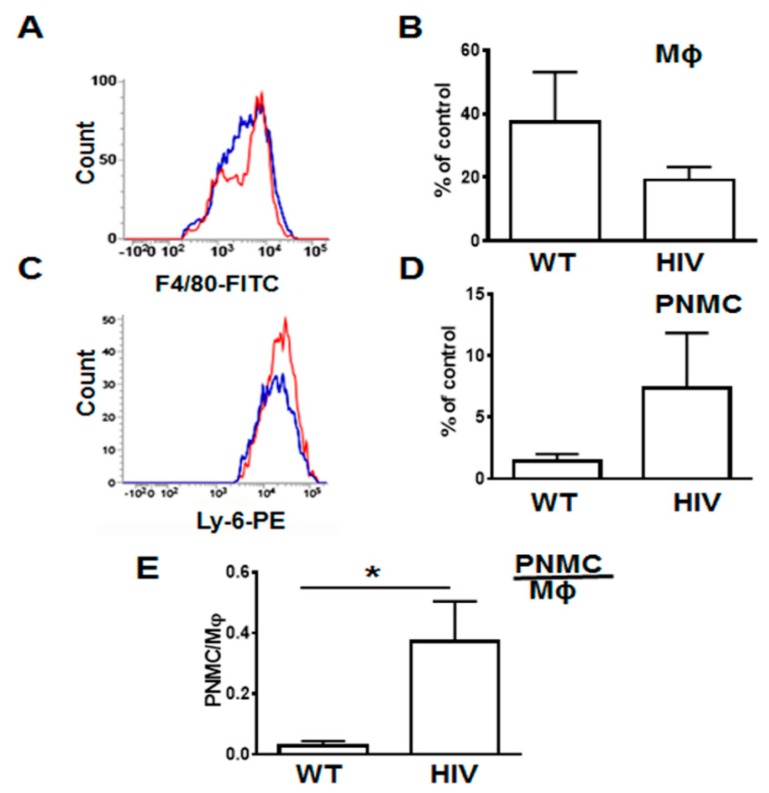
Infiltration of lung neutrophils is increased and lung interstitial macrophages is decreased in HIV-Tg mice at 24 h after LPS injection. (**A**,**B**) Representative image of flow cytometry for macrophages (A, Mϕ) and quantification of flow cytometry data (B). (**C**,**D**) Representative image of flow cytometry for neutrophils (C, PNMC) and quantification of flow cytometry data (D) in WT (blue color) and HIV-Tg (red color) mice. (**E**) neutrophil/macrophage ratios. Results were obtained from 5 mice were used for quantification. Asterisk indicates * *p* < 0.05.

**Figure 4 viruses-12-00204-f004:**
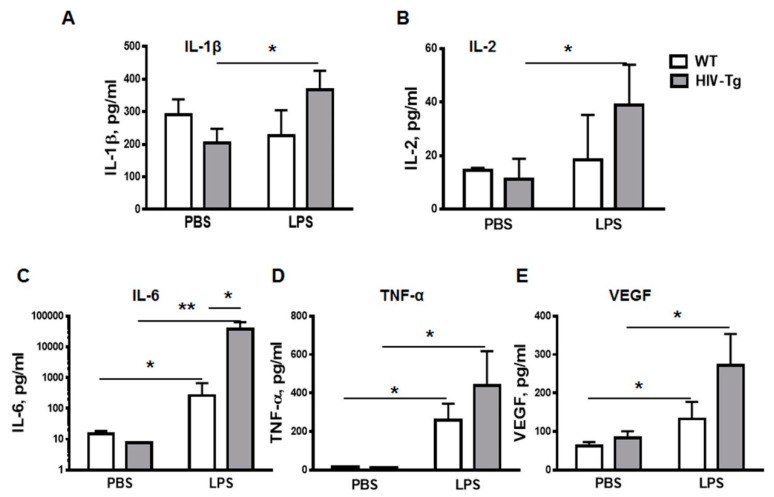
LPS-induced systemic pro-inflammatory cytokines in WT and HIV-Tg mice. Levels of circulating cytokines were measured by multiplex suspension array at 24 h after PBS and LPS administration. (**A**) IL-1β, (**B**) IL-2, (**C**) IL-6, (**D**) TNF-α. (**E**) Levels of circulating VEGF were measured by enzyme-linked immunosorbent assay (ELISA). Asterisks indicate: * *p* < 0.05 and ** *p* < 0.01.

**Figure 5 viruses-12-00204-f005:**
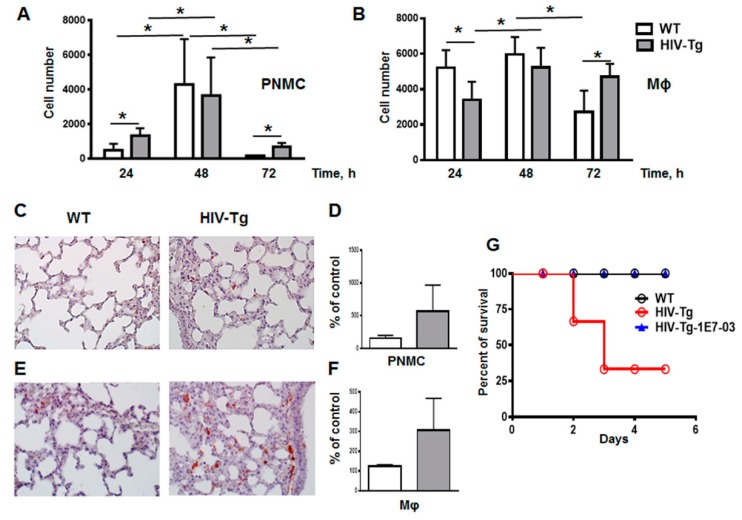
Resolution of lung inflammation is delayed in HIV-Tg mice. (**A**,**B**) Time course of resolution of lung neutrophil (A, PMNC) and macrophage (B, Mϕ) infiltration in WT (white color) and HIV-Tg (grey color) mice detected by flow cytometry. Asterisk indicates * *p* < 0.05. (**C**,**D**) Immunostaining (panel C, red color) and quantification (panel D) of lung neutrophils. Results are shown as percent of WT-PBS level. Mean and standard error are shown. Hematoxylin is used for counterstaining (purple nuclear staining). Original magnification is 400×. (**E**,**F**) Immunostaining (panel E, red color) and quantification (panel F) of lung macrophages. Results are shown as percent of WT-PBS level. Mean and standard error are shown. Hematoxylin is used for counterstaining (purple nuclear staining). Original magnification is 400×. (**G**) Survival of WT (black) and HIV-Tg mice (red) after administration of LPS and subsequent IP injection with 1E7-03 (blue). *N* = 9 for HIV-Tg mice, *N* = 10 for WT mice and *N* = 6 for HIV-Tg mice injected with 1E7-03 (HIV-Tg-1E7-03).

**Figure 6 viruses-12-00204-f006:**
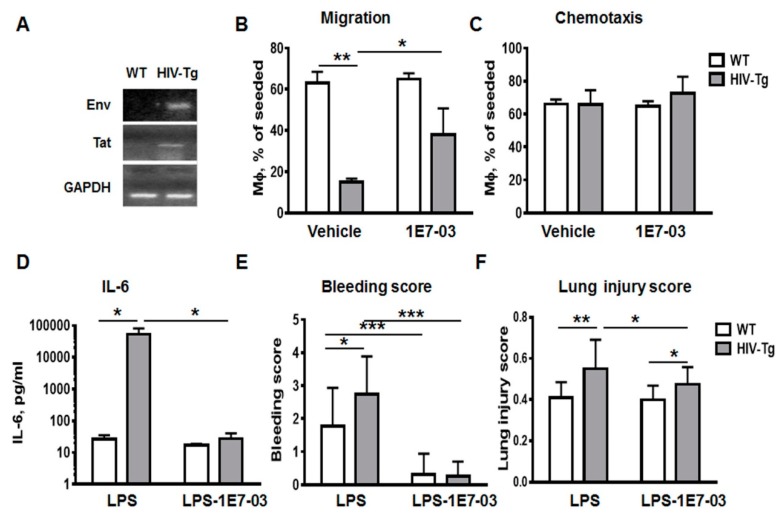
Administration of HIV transcription inhibitor improves migration of macrophages in vitro and reduces lung injury in HIV-Tg mice. (**A**) Real-time polymerase chain reaction (RT-PCR) for HIV-1 env and tat genes in the peritoneal macrophages isolated from WT and HIV-Tg mice. (B and C) Trans-endothelial migration (**B**) and chemotaxis (**C**) of peritoneal macrophages isolated from WT and HIV-Tg mice. 1E7-03 HIV-1 transcriptional inhibitor (1 μM) or 10% dimethyl sulfoxide (DMSO, vehicle) were added to macrophages. (**D**–**F**) Administration of 1E7-03 in HIV-Tg mice (1 μg/mg of body weight) significantly reduced levels of circulating IL-6 (**C**), bleeding score (**D**) and lung injury score (**E**). Asterisks indicate: * *p* < 0.05; ** *p* < 0.01 and *** *p* < 0.005.

**Figure 7 viruses-12-00204-f007:**
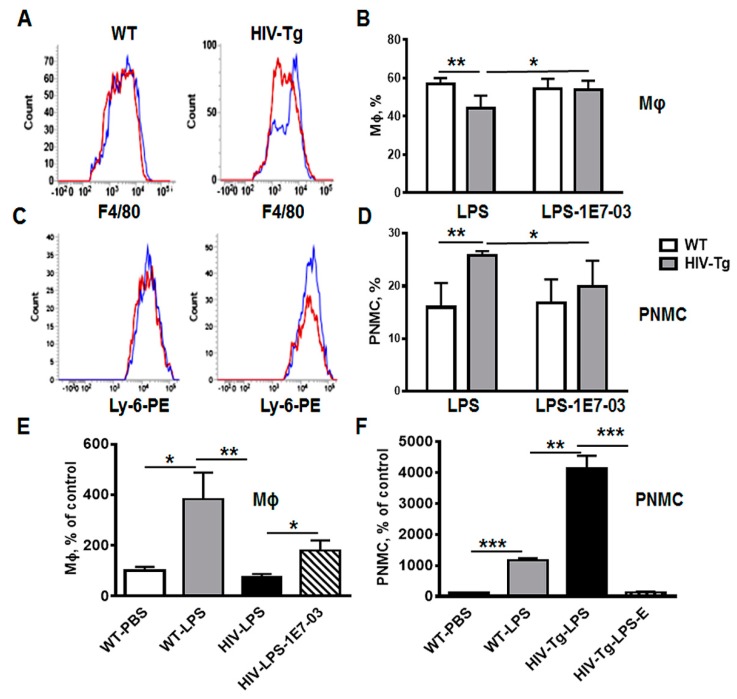
Administration of HIV-1 transcription inhibitor decreases neutrophil levels and increases macrophage levels in HIV-Tg mice injected with LPS. (**A**,**B**) Representative flow cytometry analysis of lung macrophages (A, F4/80-FITC) and neutrophils (B, Ly-6-PE) in WT and HIV-Tg mice at 48 h after LPS administration. Blue color indicates vehicle, red color—1E7-03 (HIV-1 transcriptional inhibitor). (**C**,**D**) Percent of lung macrophages (C) and neutrophils (PNMC, D) quantified by flow cytometry. Three mice were used for each treatment. Flow cytometry samples were run in duplicate. Asterisks indicate: * *p* < 0.05, ** *p* < 0.01, *** *p* < 0.005.

**Table 1 viruses-12-00204-t001:** Parameters for lung injury evaluation.

	Parameter	Score Per Field		
		0	1	2
I.	Neutrophils in the alveolar space	0	1–5	>5
II.	Neutrophils in the interstitial space	0	1–5	>5
III.	Hyaline membranes	0	1	>1
IV.	Proteinaceous debris filling air spaces	0	1	>1
V.	Alveolar septal thickening	0	2x–4x	>4x

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
