# Peer review of "HIV-1 Transcription Inhibitor 1E7-03 Restores LPS-Induced Alteration of Lung Leukocytes’ Infiltration Dynamics and Resolves Inflammation in HIV Transgenic Mice"

_viruses, 2020, doi:10.3390/v12020204_

Round 1

Reviewer 1 Report

This paper, by Dr. Jerebtsova and colleagues, describes the alteration of the type of cell filtration and cytokine expression in the lungs after LPS treatment in HIV-Tg mice, and the impact of the HIV-1 transcription inhibitor, 1E7-03. Although the authors looked at a significant issue in HIV infected patients, the data presented is not really reliable. Several important points have to be considered:

-Figure 1A: why did the PBS have an effect in HIV-Tg mice? Why did the LPS have no effect on HIV-Tg mice compared to WT ? Authors did not discuss it.

Figure 1B: statistics may be incorrect.

-Figure 2 and 3: I believe that the p values are not corresponding to the number of stars on the graphs. It would be better to show the degree of significance with the number of stars to give a better representation of the data.

-Figure 4: The detection of the cytokines should not be performed only on the serum since the different cytokines could be released by several compartments. It should also be performed on lungs cells. The expression of the oxidative stress markers and viral mRNAs would be interesting to observe as well.A similar remark for the p values as in previous figures.

Figure 5: -A similar remark for the p values as in previous figures.

              -the use of the data of the death of the mice is not interesting if the authors do not show that the mice that died are the ones that showed the worst infiltration alterations and cytokine expression. This is true especially when the sera of the mice present an alteration of the pattern of cytokines, which could be due to their release from other compartments.

Figure 6B to E: -these figures are missing the condition of the effect of the compound on WT LPS.               

                       - statistics should be verified.

                       - the expression of viral mRNA should be shown as well.

                       - it is also missing the toxicity induced by the compound 1E7-03.

-Figure 7: there were no significant differences in the number of macrophages and PNMCs between WT and HIV-Tg in absence of 1E7-E3. This makes it difficult to appreciate the effect on the number of cells when the compound is added.

-The English needs to be improved.

Reviewer 2 Report

In this manuscript, the authors demonstrate that mice made transgenic with a partial human immunodeficiency virus genome develop more protracted lung pathology when treated with intraperitoneal (IP) lipopolysaccharide (LPS).  While the IP LPS injection initially induced similar levels of systemic inflammation and pulmonary inflammation in wild type and transgenic mice, the transgenic mice showed some delay in monocyte recruitment/infiltration and delayed resolution of leukocyte infiltration relative to wild type mice. In vitro experiments showed that monocytes derived from the transgenic mice migrated across an endothelial cell barrier at a slower rate than those from the wild type mice and the authors related this migratory deficit to the delayed accumulation of monocytes in the lungs and possibly to the delayed resolution of pulmonary pathology.  The article is generally well written, but there are numerous minor inconsistencies of tense and use of plurals that detract from the quality, which could be easily edited out. Data supporting the authors' assertions was collected and presented appropriately in most figures and in the text with histology and serum analysis supporting the basic conclusions.  While this article should be of general interest to readers, the rationale for the study and interpretation of the data could be better developed in relation to relate to human HIV infection.

Firstly, in the introduction, there should be a better description of pulmonary pathology in treated HIV infection and what aspects might relate to systemic inflammation versus the presence of HIV-infected pulmonary macrophages or other pulmonary infections such as cytomegalovirus.  Secondly, the transgenic model should be much better described in terms of what HIV genes are encoded and what promoter drives their expression.  Does the LPS activate HIV transcription in this model or just increase background inflammation?  What HIV proteins are expressed?  If HIV env is expressed, does this interact with surface proteins on endothelial cells and  slow transendothelial migration.  Thirdly, the transcriptional inhibitor and its mechanism of action needs to be described in greater detail.  In addition, the potential specific meaning of individual cytokines measured and their comparison should be described more clearly. 

Minor comments

I had trouble grasping the rational interpretation underlying of the results presented in section 3.2 in relation to the authors' main points.

Line 144: At least (here)

Line 155 env(y)

Line 172 (calculates)

Line 190 (duplicates)line 212;  Student's (T)-test

Line 231 (characterized)

Line 251 2-(folds)

Line 300-301:  inconsistent abbreviation of interleukin as Il and IL

Line 345: was (it was)

Line 365: accumulation(s)

Line 383 cell(s)

Line 426; duplicate(s)

Author Response

Generate comment:

In this manuscript, the authors demonstrate that mice made transgenic with a partial human immunodeficiency virus genome develop more protracted lung pathology when treated with intraperitoneal (IP) lipopolysaccharide (LPS).  While the IP LPS injection initially induced similar levels of systemic inflammation and pulmonary inflammation in wild type and transgenic mice, the transgenic mice showed some delay in monocyte recruitment/infiltration and delayed resolution of leukocyte infiltration relative to wild type mice. In vitro experiments showed that monocytes derived from the transgenic mice migrated across an endothelial cell barrier at a slower rate than those from the wild type mice and the authors related this migratory deficit to the delayed accumulation of monocytes in the lungs and possibly to the delayed resolution of pulmonary pathology.  The article is generally well written, but there are numerous minor inconsistencies of tense and use of plurals that detract from the quality, which could be easily edited out. Data supporting the authors' assertions was collected and presented appropriately in most figures and in the text with histology and serum analysis supporting the basic conclusions.  While this article should be of general interest to readers, the rationale for the study and interpretation of the data could be better developed in relation to relate to human HIV infection.

Response: We thank the reviewer for the overall positive view of our study. We carefully edited the manuscript, to remove the inconsistencies in tense and plurals as pointed out. We also added rational to connect this study to the HIV-1 infection in humans as described below in the responses to the specific comments.

Specific Comment 1. Firstly, in the introduction, there should be a better description of pulmonary pathology in treated HIV infection and what aspects might relate to systemic inflammation versus the presence of HIV-infected pulmonary macrophages or other pulmonary infections such as cytomegalovirus. 

Response: We added the requested information to the Introduction.

Specific Comment 2. Secondly, the transgenic model should be much better described in terms of what HIV genes are encoded and what promoter drives their expression. 

Response: We agree and added this information to the Introduction.

Specific Comment 3.  Does the LPS activate HIV transcription in this model or just increase background inflammation? 

Response: LPS does not directly activate HIV transcription but induced inflammation stimulating organs infiltration of macrophages and neutrophils. We added this information to the Discussion.

Specific Comment 4.  What HIV proteins are expressed?  If HIV env is expressed, does this interact with surface proteins on endothelial cells and  slow transendothelial migration. 

Response: Seven out of nine HIV-1 proteins (excluding Gag and Pol) are expressed in this mouse model. Env is expressed in the mouse macrophages (Fig.6A), but we think that Nef might be another factor that affects migration of macrophages in HIV-Tg model. The experiments are currently performed in our lab and another manuscript is under preparation.

Specific Comment 4. Thirdly, the transcriptional inhibitor and its mechanism of action needs to be described in greater detail. 

Response: 1E7-03 is a tetrahydroquinoline-based small molecule that inhibits HIV-1 transcription with IC50=0.9 μM and does not induces cytotoxicity at concentrations below 15 μM (Ammosova et al., British J. Pharm. 2014, 171:5059-75). 1E7-03 disrupts the interaction of Tat protein with host protein phosphatase 1 (PP1). The inhibitor was further tested in vivo in  HIV-1 infected humanized mice (Lin et al., Oncotarget, 2018, 8:76749-76769).We added this information to the Material and Methods section.

Specific Comment 5. In addition, the potential specific meaning of individual cytokines measured and their comparison should be described more clearly. 

Response: Increase in circulating IL-6, TNF-α, IL-1β and C-reactive protein has been associated with chronic lung disease in HIV-1 patients (Fitzpatrick ME, AIDS 2014 and AIDS 2016; Singh S 2018 Immunol. Lett). We also tested serum levels of vascular endothelial growth factor VEGF, which is a major factor of endothelial permeability. In airways VEGF induces vascular leakage and airway edema, and enhances chemotaxis for monocytes in COPD (Bakakos P 2016 Curr Top Med Chem). Thus we evaluated these proteins levels in HIV-Tg and WT mice after LPS injection. This information is added to the Introduction and Discussion.

Minor comments

Comment 1. I had trouble grasping the rational interpretation underlying of the results presented in section 3.2 in relation to the authors' main points.

Response: We rephrased the rational for section 3.2.

Comment 2. Line 144: At least (here)

Response: here replaced by four

Comment 3. Line 155 env(y)

Response: “y” is removed

Comment 4. Line 172 (calculates)

Response: “S” is removed

Comment 5. Line 190 (duplicates) line 212; Student's (T)-test ?

Response: Line 190 (line 213 in the revised manuscript) describes flow cytometry procedure. Line 212 (revised line 235) describes statistical analysis that was updated to include the description of the methods (i.e. T-test and ANOVA). Thus the duplication was removed.

Comment 6. Line 231 (characterized)

Response: “d” is removed.

Comment 7. Line 251 2-(folds)

Response: changed to “twofold”

Comment 8. Line 300-301:  inconsistent abbreviation of interleukin as Il and IL

Response: Abbreviations were changed to IL.

Comment 9. Line 345: was (it was)

Response: ‘it was” was removed

Comment 10. Line 365: accumulation(s)

Response: “s” was removed

Comment 11. Line 383 cell(s)

Response: “s” was removed”

Comment 12. Line 426; duplicate(s)

Response: “s” was removed

Round 2

Reviewer 1 Report

In their response, the authors addressed my concerns with new data and an improved explanation of the results. However, other specific details could further improve the paper to suit Viruses journal expectations:

-The paragraph from line 43 to 54 could be removed, it does not bring anything to the story and disturbs the flow of the introduction.

-The authors should be careful in not missing some words or space between words such as “in” between "reduced" and "HIV Tg mice" in line 101, or "twofold" in line 273...

-The paragraph from line 246 to 252 could be written differently to have a better flow. For instance: “Pulmonary edema is a major complication of LPS-induced acute lung injury (43). The lung wet-to-dry ratios for non-injected HIV-Tg mice were previously shown to be higher than non-injected WT mice, indicating that the lungs of HIV-Tg mice have edema and predisposition to pulmonary complications as previously reported [37]. To confirm the edema, wet-to-dry ratios of lung tissue were assessed (Fig.1A)”.

-The following sentence could be deleted: “Because i.p. injection of PBS might stimulate inflammation in mice, we used it as a control for LPS injection. Injection of PBS did not induce lung edema either in WT or in HIV-Tg mice (not shown)”.

-The paragraph from line 258 to 263 should be rewritten to be clearer.

-Authors did not comment about the PBS vs LPS conditions for Fig.1C.

-Supplemental Figs 1A and 1B from line 275 are not present in the document.

-The paragraph from line 276 to 279 should be rewritten to be clearer.

-Fig. 3 could be organized differently:

                          -Fig.3A and Fig.3B could go to supplemental part,

                          -Fig.3C upper part should be in the same panel than Fig.3D

                          -Fig.3C lower part should be in the same panel than Fig.3E

                          -Fig.3F on the bottom.

-Authors should add “and macrophages” after leucocytes in line 315.

- The paragraph from line 318-321 could be deleted: “The alteration in lung leukocytes infiltration might be associated with reduced level of inflammation in HIV-Tg mice, as the mice express seven of nine HIV-1 proteins that might potentially alter inflammatory response after LPS administration.”

-The paragraph from line 394 to 399 should be rewritten to be clearer and include which figure number the authors are referring to.

-Authors should put together Fig.7 and supplemental Fig.2.: Fig.7A and C should be put in the upper panel, Fig.7B and D in the lower panel then include supplemental Fig.2  as well.

-It would be interesting if the authors add a paragraph about oxidative stress and nitrosative stress in the discussion as future work to be investigated.

Author Response

Reviewer 1.

General Comment: In their response, the authors addressed my concerns with new data and an improved explanation of the results. However, other specific details could further improve the paper to suit Viruses journal expectations:

Response: We thank the reviewer for the positive assessment of our manuscript. Please see below point-to-point responses to the remaining comments.

Specific Comment 1. The paragraph from line 43 to 54 could be removed, it does not bring anything to the story and disturbs the flow of the introduction.

Response: This paragraph was added in the first round of revision in response to the Specific Comment of the reviewer 2: “Firstly, in the introduction, there should be a better description of pulmonary pathology in treated HIV infection and what aspects might relate to systemic inflammation versus the presence of HIV-infected pulmonary macrophages or other pulmonary infections such as cytomegalovirus.”  We do agree that this added paragraph interrupts the flow, so we removed it.

Specific Comment 2: The authors should be careful in not missing some words or space between words such as “in” between "reduced" and "HIV Tg mice" in line 101, or "twofold" in line 273...

Response: We corrected these misspells.

Specific Comment 3: The paragraph from line 246 to 252 could be written differently to have a better flow. For instance: “Pulmonary edema is a major complication of LPS-induced acute lung injury (43). The lung wet-to-dry ratios for non-injected HIV-Tg mice were previously shown to be higher than non-injected WT mice, indicating that the lungs of HIV-Tg mice have edema and predisposition to pulmonary complications as previously reported [37]. To confirm the edema, wet-to-dry ratios of lung tissue were assessed (Fig.1A)”.

Response: We added this correction (lane 238).

Specific Comment 4: The following sentence could be deleted: “Because i.p. injection of PBS might stimulate inflammation in mice, we used it as a control for LPS injection. Injection of PBS did not induce lung edema either in WT or in HIV-Tg mice (not shown)”.

Response: We deleted this sentence.

Specific Comment 5: The paragraph from line 258 to 263 should be rewritten to be clearer.

Response: We rewrote the paragraph as follows (lane 245): “LPS injection induced bleeding in both HIV-Tg and WT mice, but no differences in bleeding scores were found”.

Specific Comment 6: Authors did not comment about the PBS vs LPS conditions for Fig.1C.

Response:  We add clarification for Fig1C as follows (lane 250): “LPS injection induced lung injury in both HIV-Tg and WT mice, but no differences were found in lung injury scores between WT and HIV-Tg mice injected with LPS (Fig.1C, WT-LPS versus HIV-LPS, p=0.83).”

Specific Comment 7: Supplemental Figs 1A and 1B from line 275 are not present in the document.

Response:  We added the requested supplemental figures (lane 263, supplemental Figs. A2A and A2B).

Specific Comment 8: The paragraph from line 276 to 279 should be rewritten to be clearer.

Response: we rewrote the paragraph as follows (lane 265): ”In contrast, in LPS-injected HIV-Tg mice there were much less infiltrated interstitial macrophages compared to WT mice, suggesting (Figs.2E and 2F, WT versus HIV-Tg, red staining, p=0.0057). “

Specific Comment 9: -Fig. 3 could be organized differently:

                          -Fig.3A and Fig.3B could go to supplemental part,

                          -Fig.3C upper part should be in the same panel than Fig.3D

                          -Fig.3C lower part should be in the same panel than Fig.3E

                          -Fig.3F on the bottom.

Response: we revised the figure as requested. Panels 3A and 3B were moved to Supplemental Figure A3 and the rest of the panels were rearranged.

Specific Comment 10: Authors should add “and macrophages” after leucocytes in line 315.

Response: we added “and macrophages” as requested.

Specific Comment 11:  The paragraph from line 318-321 could be deleted: “The alteration in lung leukocytes infiltration might be associated with reduced level of inflammation in HIV-Tg mice, as the mice express seven of nine HIV-1 proteins that might potentially alter inflammatory response after LPS administration.”

Response: We agree and deleted it.

Specific Comment 12. The paragraph from line 394 to 399 should be rewritten to be clearer and include which figure number the authors are referring to.

Response: we revised paragraph figure as requested with the reference to the Figure 2G and H.

Specific Comment 13. Authors should put together Fig.7 and supplemental Fig.2.: Fig.7A and C should be put in the upper panel, Fig.7B and D in the lower panel then include supplemental Fig.2 as well.

Response: we revised the figure as requested.

Specific Comment 14. It would be interesting if the authors add a paragraph about oxidative stress and nitrosative stress in the discussion as future work to be investigated.

Response: we added the paragraph (lanes 562-5565) to the discussion.

Reviewer 2 Report

The authors have appropriately addressed the concerns I raised in my previous review.

Author Response

Thank you for the positive accessment of our manuscript.